# Biological Activities of Bismuth Compounds: An Overview of the New Findings and the Old Challenges Not Yet Overcome

**DOI:** 10.3390/molecules28155921

**Published:** 2023-08-07

**Authors:** Jânia dos Santos Rosário, Fábio Henrique Moreira, Lara Hewilin Fernandes Rosa, Wendell Guerra, Priscila Pereira Silva-Caldeira

**Affiliations:** 1Department of Chemistry, Centro Federal de Educação Tecnológica de Minas Gerais, Belo Horizonte 30421-169, MG, Brazil; 2Institute of Chemistry, Universidade Federal de Uberlândia, Campus Santa Mônica, Uberlândia 38400-142, MG, Brazil

**Keywords:** bismuth, antimicrobial activity, antiviral activity, bacterial resistance, cancer, *Helicobacter pylori*, antiparasitic activity

## Abstract

Bismuth-based drugs have been used primarily to treat ulcers caused by *Helicobacter pylori* and other gastrointestinal ailments. Combined with antibiotics, these drugs also possess synergistic activity, making them ideal for multiple therapy regimens and overcoming bacterial resistance. Compounds based on bismuth have a low cost, are safe for human use, and some of them are also effective against tumoral cells, leishmaniasis, fungi, and viruses. However, these compounds have limited bioavailability in physiological environments. As a result, there is a growing interest in developing new bismuth compounds and approaches to overcome this challenge. Considering the beneficial properties of bismuth and the importance of discovering new drugs, this review focused on the last decade’s updates involving bismuth compounds, especially those with potent activity and low toxicity, desirable characteristics for developing new drugs. In addition, bismuth-based compounds with dual activity were also highlighted, as well as their modes of action and structure–activity relationship, among other relevant discoveries. In this way, we hope this review provides a fertile ground for rationalizing new bismuth-based drugs.

## 1. Introduction

Historically, the use of bismuth as a medicine is old and bismuth salts have been utilized for treating skin lesions and syphilis for more than 300 years. For instance, bismuth subnitrate was used empirically in the treatment of dyspepsia by Louis Odier in 1786 [1]. Despite a decline in its use due to the emergence of antibiotics, discoveries showcasing its effectiveness against *Helicobacter pylori* (*H. pylori*) have led to a resurgence of bismuth in modern medicine [2].

Although a heavy metal, bismuth is non-toxic and has no carcinogenic effect [3]. Thus, its safety and efficacy have attracted the attention of researchers and the pharmaceutical industry, which has led to the development of several drugs, some of which have already been successfully incorporated into clinical practices, such as the examples shown in Figure 1 and Table 1.

In addition to treating gastrointestinal diseases and microbiological infections, such as syphilis, diarrhea, gastritis, and colitis [9,10,11], bismuth compounds exhibit potent antimicrobial activities against Gram-positive and Gram-negative bacterial pathogens [1], *Leishmania* [12], and fungi [13]. Recently, the pharmacological proprieties of bismuth compounds have been extended, demonstrating their potential for treating multidrug-resistant microbial infections as an antibiotic adjuvant [14,15], cancer, and viral infection, highlighting its promising ability to suppress SARS-CoV-2 [16,17,18]. The advancement of bismuth beyond the fight against *H. pylori* shows that the development of drugs based on bismuth is an attractive field and that there is still much fertile ground to be explored, considering its safety and pharmacological potential.

Despite the success of bismuth in modern medicine, its low bioavailability in physiological environments represents a challenge that must be faced. Thus, many bismuth compounds with improved physicochemical properties, as well as new administration modalities for bismuth release have been recently prepared and biologically tested. In light of this, the present review explores the recent progress in bismuth-based compounds, particularly in developing more effective drugs with new biological applications, improved bioavailability, potent activity, and low toxicity, as well as those with dual or multiple biological activities, including, when possible, their mechanisms of action and structure–activity relationship.

## 2. Bioavailability Challenges Associated with Bismuth-Based Compounds

The effectiveness of a drug is determined by its bioavailability, which refers to the amount of the active ingredient that is absorbed and available to produce systemic effects [19]. However, commercially available bismuth compounds such as BSS, CBS, and RBC are not readily soluble in the stomach’s acidic environment (pH 2), leading to poor absorption of Bi^3+^ ions into the bloodstream. This results in limited bioavailability, with less than 1% of orally ingested bismuth being absorbed and quickly excreted in the urine [20]. Thus, studies have shown that even with multiple doses, the maximum plasma concentration of bismuth remains below the toxic threshold of 50 μg L^−1^ [21,22,23]. Even during treatment, pathogens are only exposed to small amounts of bismuth for short periods due to its low solubility, low absorption rate, and rapid clearance from the body. Mammalian cells have efficient uptake and detoxification mechanisms of Bi^3+^ ions, which are mediated by glutathione (GSH) and multidrug-resistant protein (MRP) [24]. The stable conjugate Bi-GSH is transported into perinuclear vesicles by MRP or eliminated through MRP at the cellular level [25,26].

The solubility of a compound in polar solvents is crucial for its biological application. Other factors that affect a compound’s bioavailability, efficacy, and dosage include its molecular structure, intermolecular interactions, and stability as an active pharmaceutical ingredient. Despite being used for a century, determining the molecular structures of commercial bismuth compounds—the specific arrangement of atoms—is challenging. For instance, it was only recently that the structure of BSS was discovered using modern three-dimensional electron diffraction techniques (Figure 1) [5]. In the same direction, the structure and formula of bismuth subgallate were elucidated by continuous rotation electron diffraction only in 2017 (Figure 1) [4]. The structure of CBS, another bismuth-containing pharmaceutical, was identified only after prolonged use [27]. These studies revealed the structures of the bismuth-containing pharmaceuticals as coordination polymers, providing insight into the formula, poor solubility, and acid resistance of these compounds. Interestingly, some reports describe other drugs’ influence on bismuth bioavailability. For example, some medications, such as omeprazole, could increase the bismuth blood level [28].

Considering the above discussions, many researchers have explored several approaches to improve the bioavailability of bismuth in physiological environments. In this regard, this review sheds light on two approaches: developing novel bismuth compounds (Section 3) and incorporating bismuth-based agents into various devices for controlled release (Section 4).

## 3. Pharmacological Properties of Bismuth-Based Compounds

### 3.1. Antibacterial Activity

Worldwide, the continuous flow of drug-resistance responses developed by some bacteria and the spread of multidrug-resistant (MDR) pathogens threaten the effective prevention and treatment of common bacterial infections and represent a concern for global public health [29]. Certain agents containing bismuth have displayed significant potential in combatting Gram-positive and Gram-negative bacterial pathogens. In addition, some bismuth agents have also demonstrated synergistic effects when combined with antibiotics, including in cases where antibiotic resistance had previously occurred [14,30].

#### 3.1.1. Modes of Action against *H. pylori* and Bacterial Resistance Mechanisms

*H. pylori* is a transmissible Gram-negative pathogen that infects the stomach of half of the human population worldwide, causing chronic gastritis and peptic ulcers [31], as Barry Marshall and Robin Warren discovered in 1982 [32]. In addition, this bacterium is also associated with gastric cancer [22]. Due to the effective treatment of *H. pylori*-associated gastrointestinal ailments, nowadays co-administrated with antibiotics and a proton pump inhibitor, the antiulcer activity of bismuth-based agents has remained in clinical use for decades, although the mechanisms of action were not completely elucidated.

It is known that bismuth-based drugs form a protective coating on the ulcer crater, which prevents the erosion generated by gastric acid and contributes to the healing of the lesion [10,33]. In addition, some studies indicated that the biological targets of the antiulcer activity of bismuth are related to the oxidative stress of *H. pylori* [34]. Bismuth also can inactivate enzymes involved in respiration, such as F1-ATPase [35], and other enzymes, such as urease [21,36] and alcohol dehydrogenase [37], besides interfering with a range of Zn^2+^- and Fe^3+^-regulating proteins [38]. However, recently, through bioinformatics analysis in combination with bioassays, some investigations have provided novel and deeper insights into the complex molecular multi-targeted mode of action by which bismuth eradicates *H. pylori* [39] (Figure 2). For example, Sun and collaborators demonstrated that bismuth drugs disrupted multiple essential pathways in the pathogen, including damaging the oxidative defense systems and eliminating the bacterial pH-buffering ability by the binding and functional perturbation of several key enzymes [40]. They also discovered a new target (HpDnaK) of bismuth drugs to inhibit bacterium–host cell adhesion [41]. The systemic responses of six *H. pylori* strains isolated from patients cultured with and without bismuth were recently explored by Yao et al. through integrative proteomic and metabolomic analyses [42]. They found that bismuth inhibited *H. pylori* growth in vitro via the downregulation of virulence proteins CagA and VacA, the disruption of the flagella assembly responsible for bacterial colonization, and the inhibition of antioxidant enzymes. Furthermore, they observed that diverse metabolic pathways related to growth and RNA translation and biological processes in *H. pylori*, including antioxidant response and purine, pyrimidine, amino acid, and carbon metabolism, were disrupted by bismuth [42].

While most proteins interact with only a few others, a small number of proteins, termed hubs, are core proteins highly connected with other proteins [43]. Bismuth can bind *H. pylori* hub proteins [41]; thus, bismuth is expected to be more lethal to this pathogen and less susceptible to developing resistance. In addition, bismuth also has a multifactorial mechanism of action against *H. pylori*; therefore, the chances of the pathogen developing resistance linked to molecular multi-target agents are lower. However, despite bismuth compounds having potential against *H. pylori*, their eradication rate is only 10% to 20% when administrated alone [44]. These limited bacterial eradication rates by bismuth agents can be justified by the *H. pylori* defense mechanisms employed to escape bismuth toxicity (Figure 2). Recently, Kumar et al. reported a study indicating novel defense mechanisms of *H. pylori* against bismuth [45]. They demonstrated that even low doses of bismuth prompt profound changes in *H. pylori* physiology, inducing the formation of homogeneously sized membrane vesicles with unique protein cargo content enriched in bismuth-binding proteins. These vesicles are responsible for mediating the bismuth expulsion from the bacteria cell. Additionally, a probable local DNA protective response associated with polyphosphate granule formation induced by bismuth exposure was observed. The poly-P granules cause bacteria nucleoid compaction, protecting it from bismuth toxicity [45].

As to developing new bismuth drugs to combat *H. pylori*, some new Bi^3+^ compounds exhibiting significative anti-*H. pylori* potential were reported in the last decade. For example, a binuclear Bi^3+^ complex containing pyridine 2,6-dicarboxylate and thiourea was recently prepared and assessed against three strains of *H. pylori*, displaying excellent activity with MIC (minimum inhibitory concentration) values of 8.84 μM, about two and four times more active than CBS and BSS, respectively [46]. In a similar direction, Kowalik et al. prepared two Bi^3+^ coordination polymers with the same pyridine-2,3-dicarboxylate ligand. One presented MIC values of 11.4 and 13.4 μM against two *H. pylori* strains, 26695 and N6, respectively, being about twice more active than BBS in the same bacteria strains [47]. Notably, commercially available medicinal bismuth carboxylates BSS, RBC, and CBS have MIC values against *H. pylori* ranging from 8 to 12.5 µg mL^−1^ [48].

In the last decade, the research group of Professor Andrews reported several works containing Bi^3+^ complexes with various ligands, in which many exhibited significative activity against three *H. pylori* strains (26695, B128, and 251) [49,50,51,52,53]. Additionally, Andrews and colleagues have conducted various studies on the relationship between the structures of Bi complexes and their antimicrobial activity. One important aspect they have highlighted is the role of the type and coordination mode of the ligand to bismuth, including donor atoms, hapticity, and the influence of substituents. The physicochemical properties of the final complex formed are crucial to its biological action, as evidenced by their research and others [54]. Sun and colleagues also comprehensively examined bismuth-based compounds as antimicrobial agents. Their study focused on identifying the molecular targets of bismuth agents and their mechanisms of action as antimicrobial agents [1].

#### 3.1.2. Antibacterial Activity beyond *H. pylori* and Reversion of Bacterial Drug Resistance

Bismuth is not only active against *H. pylori*; for example, Bi compounds with a broad spectrum of activity against Gram-negative and/or Gram-positive bacteria, including antibiotic-resistant bacterial lines, have been continually reported. Andrews et al. reported a series of homoleptic and heteroleptic Bi^3+^ flavonolate complexes active against some Gram-positive and -negative bacterial lines at the micromolar level, including methicillin-resistant *Staphylococcus aureus* (MRSA) and vancomycin-resistant *Enterococcus* (VRE). It was also observed that the heteroleptic complexes exhibited effective antibacterial activity at micromolar levels, although they did show some level of toxicity towards mammalian cells. At the same time, the homoleptic complexes showed lower toxicity to mammalian cells; however, their antibacterial activity was also lower [55]. Subsequently, they evaluated the structure–activity relationships in di-aryl Bi phosphinates on antibacterial activity and demonstrated that the Bi^3+^-bound aryl group profoundly influences activity. Indeed, some complexes have the most significant activity towards MRSA and VRE in the range of 0.63 to 1.25 μM [56]. In another report, they also observed that a series of phenyl bis-phosphinate Bi^3+^ complexes exhibited effective antibacterial activity against MRSA and VRE [57]. The most active (**1**) is shown in Figure 3. Furthermore, they reported a series of novel diphenyl mono-phosphinate Bi complexes that exhibited effective antibacterial activity towards various bacteria, including MRSA and VRE, although these complexes also show some toxicity toward mammalian cells [58]. The above results indicate that exploring bismuth as a source for new antimicrobial agents is challenging, but it could result in the development of novel antibiotics to combat critical pathogens in clinical settings.

Tuberculosis, a poverty-related disease, is also considered neglected in drug research and development. Moreover, there is the emergence of resistant strains to all drugs available for treatment. For example, clinical isolates resistant to isoniazid and rifampicin, plus any fluoroquinolone and at least one of the four injectable second-line drugs (streptomycin, for instance) have already been reported in many countries [59], which makes necessary the search for novel compounds effective against *Mycobacterium tuberculosis*. In this sense, Machado et al. synthesized an octahedral Bi^3+^ complex containing the ligand pyridine-2-thiol-1-oxide (**2**) that showed excellent activity, both on the standard strain H37Rv ATCC 27294 (pan-susceptible) and in five clinical isolates resistant to the standard drugs isoniazid and rifampicin, displaying MIC values in the range of 2.49 to 21.18 µM [60].

Indeed, some studies have proven that bismuth-based combination therapy can effectively treat infections caused by antibiotic-resistant bacteria [1]. For instance, diseases caused by *Enterobacteriaceae* producing metallo-β-lactamases are hard to treat. Therefore, searching for effective metallo-β-lactamase inhibitors to restore the effectiveness of existing antibiotics is highly desirable. Although these inhibitors are not clinically available to date, a recent study has demonstrated a high potential of bismuth-based drugs to inhibit metallo-β-lactamase enzymes and restore the in vivo efficacy of the antibiotic meropenem [15]. Furthermore, the ability of bismuth to inhibit metallo-β-lactamase activity suggests that bismuth-based drugs could be repurposed alongside clinically used antibiotics as a co-therapy to address the current antibacterial resistance crisis [61]. In other studies, Bisceglie and collaborators tested the synergetic effect of a series of Bi^3+^ thiosemicarbazone complexes with the antibiotic meropenem in combination therapy in carbapenem-resistant *Klebsiella pneumoniae* carrying the NDM-1 gene, which is responsible for the production of the correspondent metallo-β-lactamase [62]. One of the complexes (**3**), with negligible antibacterial activity (MIC >250 μM) when used in monotherapy, displayed synergic activity with meropenem, restoring antibiotic sensitivity in the strain-producing NDM-1 enzyme [62]. Interestingly, the authors reported that complexes bearing a hydroxyl moiety involved in the coordination of the bismuth, *O*,*N*,*N*′,*S*-chelating mode, did not show any relevant antimicrobial activity and also didn’t displace Bi^3+^ to Zn^2+^ in a spectrophotometric titration experiment [62].

Deng and collaborators recently reported another successful example of the capacity of bismuth drugs to reverse resistance and repurpose clinically approved antibiotics [14]. They found that five bismuth drugs, commonly used in the clinical treatment of stomach-associated diseases, effectively boost the antibacterial activity of tigecycline against the gene Tet(X), responsible for conferring tigecycline resistance in Gram-negative bacteria. Furthermore, mechanistic studies showed that bismuth drugs effectively suppress the enzymatic activity of the Tet(X) resistance protein [14].

Periodontitis is a gum disease that could eventually lead to tooth loss and is an increased risk factor for several systemic conditions, such as cardiovascular and pulmonary diseases [63]. *Porphyromonas gingivalis* is the key pathogen of this oral disease. This bacterium can co-aggregate with other species, forming biofilms in the inflamed gingival tissues [64] and resisting common antimicrobials such as metronidazole [65]. Sun, Jin, et al. conducted a comprehensive investigation indicating that bismuth drugs can suppress this bacterium in its planktonic, biofilm, and internalized states [66]. The study revealed that bismuth upregulated iron metabolism and stress response while downregulating the energy metabolism of *P. gingivalis*. Furthermore, they demonstrated that bismuth perturbs the expression of a bacterium’s genes, translation, and virulence beyond to reconcile the immuno-inflammatory responses of *P. gingivalis*-invaded human cells [67].

Other research groups, including ours, have adopted a strategy of coordinating Bi^3+^ with antibiotics or bioactive molecules because bismuth has been found to exhibit synergistic action with antibiotics, which may help reverse antibiotic resistance or obtain compounds with other pharmacological activities [67,68,69,70,71,72].

### 3.2. Antifungal Activity

According to Frei et al., there are currently less than ten antifungal drugs in clinical development, a serious concern as new fungal strains resistant to most current antifungals rapidly spread worldwide [73]. Besides the resistance, the usefulness of antifungals is also limited by their narrow spectrum of activity and high toxicity. To make matters worse, developing antifungals has always been challenging since fungi and their human host are eukaryotes, making it difficult to identify unique targets for antifungal drugs [74]. To overcome the resistance phenomenon and prevent a more serious public health situation, new effective antifungals are needed, especially if they present new mechanisms of action.

Since the antibacterial activity of bismuth compounds is well-established, several research groups have naturally prepared bismuth compounds and evaluated their efficacy against fungi [13,75,76,77,78,79,80,81,82]. For instance, three Bi^3+^ complexes, **4**–**6**, Figure 4, were tested against *Candida albicans*: **5** (MIC: 44 µM) showed better activity than fluconazole (MIC: 59 µM), the drug employed as a positive control, while the two other complexes were less active [83]. These results suggest that the para-chloro substituent’s presence on the ligand’s structure favored the antifungal activity. Subsequently, the same research group prepared a Bi^3+^ complex containing clioquinol (**7**), an antifungal and antiprotozoal drug. Complex **7** was 70-fold more active than fluconazole and approximately 3-fold more active than the its free ligand against *Candida albicans* [70].

As to the coordination compounds, a series of eight heteroleptic triorganobismuth(V) carboxylates were prepared and evaluated for their antifungal activity against four fungal strains, namely *Aspergillus fumigatus* (FCBP-66), *A. niger* (FCBP-0198), *Mucor* species (FCBP-0300), and *Aspergillus flavus* (FCBP-0064). Complexes **8**, **9**, and **10** exhibited significant antifungal activity against *A. flavus* with MIC values of 6.25 mg mL^−1^, comparable to the reference drug(s) [84].

Nanomaterials are an alternative to overcome antimicrobial resistance. An interesting study has revealed the promising properties of bismuth oxide nanoparticles (Bi_2_O_3_ NPs) in treating *Candida albicans* fungal infections and preventing biofilm formation. Notably, Bi_2_O_3_ NPs have shown better efficacy than commonly used antifungal agents such as chlorhexidine, nystatin, and terbinafine, known for their oral antiseptic and commercial antifungal properties [85].

In the last decade, several studies have supported the potential antifungal properties of bismuth-based nanomaterials [86,87,88,89,90,91]. For instance, the antimicrobial properties of BiVO_4_ nanorods were explored toward several fungal strains, and remarkable antifungal activity was observed for *Fusarium solani*. Additionally, cell culture experiments showed that BiVO_4_ was not able to induce hemolysis at low concentrations, which denotes its compatibility and low toxicity [92]. In other studies, bismuth nanoparticles (BiNPs), when tested under planktonic conditions, exhibited strong antifungal effects (MIC values ranging from 1 to 4 µg mL^−1^) against several strains of *Candida auris*, an emergent multidrug-resistant pathogenic yeast [93]. More recently, it was reported that green and non-toxic bismuth sulphide@graphitic carbon nitride (Bi_2_S_3_@g-C_3_N_4_) nanosheets (NCs) were synthesized and assessed for their antimicrobial activity. Bi_2_S_3_@g-C_3_N_4_ NCs exhibited high antifungal activity against *C. albicans*, with disinfection rates of 99.92% [94].

Despite the outstanding results shown above, it is surprising that bismuth compounds, which have proven effective against several bacterial strains, such as *H. pylori*, have not been extensively studied for their potential as antifungal agents. This seems to represent a missed opportunity.

### 3.3. Antiparasitic Activity

Several neglected tropical diseases (NTDs) are caused by parasites, such as leishmaniasis and Chagas disease. According to the World Health Organization (WHO), NTDs are prevalent in tropical areas, primarily affecting the poorest populations, especially women and children. Unfortunately, more than one billion people from developing countries face damaging health, social, and economic issues caused by NTDs [95].

Leishmaniasis is an endemic tropical disease caused by parasites of the *Leishmania* species, transmitted to humans and other species of mammals primarily via sandfly bites [40,41,42]. Leishmaniasis is mainly treated with antimony-based drugs, such as meglumine antimoniate (MA; Glucantime^®^), which have severe side effects and may not achieve clinical and parasitological cures [96]. Bismuth and antimony exhibit similar biological chemistry, which suggests that bismuth-based compounds may be a safe alternative to antimonial drugs and, in this regard, should be widely investigated. However, only a few classes of bismuth compounds have been designed and evaluated against leishmaniasis [12] and some of these promising compounds are depicted in Figure 5 and Figure 6.

The first bismuth compounds assessed for anti-leishmanial activity were prepared by Andrews and coworkers using NSAIDs (non-steroidal anti-inflammatory drugs) and substituted benzoic acids as ligands [97]. In this work, bismuth complexes bearing substituted benzoic acids showed significant anti-leishmanial activity against the promastigotes of *L. major* V121. Although these metal complexes demonstrated generic toxicity, this work encouraged further studies [68,98,99,100,101,102]. Thus, subsequently, Demicheli et al. reported a Bi^3+^ complex containing dppz (dipyrido [3,2-a:2′,3′-c]phenazine) (**11**), highly active against the promastigote form of Sb^3+^-sensitive and -resistant *Leishmania infantum chagasi* and *Leishmania amazonensis* strains. Furthermore, it was found that chelation decreased the lipophilicity of dppz, suggesting the role of this factor in improving the anti-leishmanial activity of the complex [99]. The fact that this complex has demonstrated potent activity and selectivity against resistant strains is a significant finding due to the increased clinical resistance involving antimony-based drugs, which can lead to failure in treating leishmaniasis [103].

Over the past decade, some reports have supported the promising anti-leishmanial activity of bismuth compounds [52,104,105] and some outstanding results have been presented. For example, two Bi^3+^ complexes with glycolic acid (**12** and **13**) were assessed against promastigote and amastigote forms of Leishmania, as well as human fibroblasts, and both exhibited good activity and a high degree of selectivity (13.2 for **9** and 53.9 for **10**) [106].

Two homo- and heteroleptic Bi^3+^ thiazole–thiolate complexes, **14** and **15,** were evaluated against *Leishmania major* promastigotes. **14** showed the highest activity (IC_50_ value 0.11 μg mL^−1^), and it had no toxic effect on human fibroblast cells. **14** was approximately 37 times more effective than **15**, demonstrating the phenyl group’s importance in biological activity [107]. It was also reported that seven new Bi^3+^ complexes containing ligands derived from indole-carboxylic acids had been prepared, and that two of them, **16** and **17**, exhibited comparable activity to Amphotericin B against the parasite *Leishmania major*, without any toxicity towards the mammalian cells at their effective concentration (0.19–0.39 µM) [108]. Furthermore, as reported firstly by A. Luqman et al. [107], the importance of the phenyl group coordinated with bismuth in the biological activity was also demonstrated, since the complexes containing only indole-carboxylates were not active (IC_50_ > 100 µM) against *Leishmania major*.

Some Bi^5+^ compounds, in Figure 6, notably have relevant anti-leishmanial activity. Thus, many tris-tolyl Bi^5+^ di-carboxylate complexes involving either ortho-, meta-, or para-substituted tolyl ligands were evaluated for their antiparasitic activity against *L. major* promastigotes and cytotoxicity against human fibroblasts. Two of these complexes showed significant toxicity towards promastigotes at low concentration and high selectivity indices (SI). For example, complexes **18** and **19** showed SI equal to 114 and 838, respectively. It was also demonstrated that the best activity and selectivity are observed with complexes containing o- and m-tolyl ligands, and it appears the primary influence on fibroblast toxicity is the aryl group, while the carboxylate influences promastigote toxicity [109]. Undoubtedly, these findings are noteworthy and motivate further studies.

Furthermore, Andleeb et al. evaluated a series of eight heteroleptic triorganobismuth(V) biscarboxylates (**20**–**27**) for their anti-leishmanial potential against *Leishmania tropica* KWH23 strains. The low IC_50_ values (<1 µM) revealed that the complexes are highly active, except for **25** (IC_50_ = 20 µM). Additionally, the cytotoxicity profile showed that the tris(tolyl) derivatives presented lower toxicity against isolated lymphocytes with higher anti-leishmanial potential. Finally, stability studies confirmed that Bi^5+^ complexes are stable in solution and in leishmanial culture M199. This report is fascinating due to the lack of stability involving many Bi^5+^ compounds reported in the literature [110]. Moreover, these complexes exhibited good activity and need further exploration, even as a starting point for designing new Bi^5+^ complexes.

Although the mechanism of action of bismuth on the leishmanial parasite is not yet known, Bi^5+^ compounds presumably act as prodrugs exhibiting their activity through the reductive pathway to generate the more active Bi^3+^ analog compounds [54]. In addition, the Bi^5+^ bio-reduction process through redox-active biomolecules, such as GSH and trypanothione, decreases the microbes’ oxidative stress protection, leading to the increased production of reactive oxygen species (ROS) and consequent cell death [54].

As noted above, despite the recent increase in the search for effective bismuth compounds against leishmaniasis, the results have emerged from just a few research groups. However, although there are a small number of manuscripts in the literature, the outcomes reveal that bismuth compounds are promising for treating Leishmania and must be continuously prepared and tested, including in vivo tests.

Chagas disease or American trypanosomiasis is an endemic parasitosis caused by the protozoan *Trypanosoma cruzi*, affecting millions worldwide. The primary concern is that the two drugs available to treat this illness, nifurtimox and benznidazole, exhibit severe side effects such as neuropathy, anorexia, and weight loss, among others. Add to this the fact that there is no effective treatment for the chronic form of the disease, whose symptoms include mega disease of the esophagus or colon, and cardiomyopathy, the latter being the most common cause of death [111,112]. Furthermore, there is concern that strains resistant to these drugs have already been reported [113]. Given this scenario, searching for new anti-*Trypanosoma cruzi* drugs is prudent, mainly for the chronic phase.

Beraldo and coworkers prepared two Bi^3+^ 8-hydroxyquinoline-derived hydrazone complexes, **28** and **29**, Figure 7, evaluated against *Trypanosoma cruzi*. Both complexes were more potent than benznidazole as antiparasitic agents. Complex **29** exhibited the highest activity against the extracellular parasites (EC_50_ = 0.06 µM) and the highest anti-*T. cruzi* activity against the amastigote form (EC_50_ = 2.31 µM). The mode of action of **29** on trypomastigotes was investigated using annexin V-fluorescein isothiocyanate (FITC) and propidium iodide (PI), and the results suggested the occurrence of cell death by necrosis [114]. The pioneering results by Beraldo et al. [114] indicate that Bi^3+^ complexes with 8-hydroxyquinoline-derived hydrazones are a starting point for developing new bismuth compounds effective against trypanosomes and deserve attention.

The antiparasitic effects of bismuth lipophilic nanoparticles (BisBAL NPs) against *Trichomonas vaginalis* were recently evaluated. The results showed that after 24 h of exposition to 500 µg mL^−1^ of BisBAL NPs, the *Trichomonas vaginalis* growth was inhibited more than 90%, with efficacy similar to metronidazole at 1.3 µg mL^−1^, the reference drug [115,116]. This finding is noteworthy because *Trichomonas vaginalis* is the most common vaginal parasitosis in women worldwide, with about 5% of the trichomoniasis clinical cases being caused by parasites resistant to metronidazole. In this sense, it is important to highlight that so far, no resistance to bismuth has been reported [21].

### 3.4. Antitumoral Activity

Due to the growing interest in medicinal inorganic chemistry since the emergence of cisplatin as a chemotherapeutic, several non-platinum metal complexes with a pivotal role in treating various diseases have been developed or improved/rationalized from old drugs [117,118]. Aiming to overcome some limitations of platinum-based drugs, such as acquired resistance and severe side effects, many bismuth compounds have been increasingly investigated as alternative anticancer agents, since they generally exhibit low toxicity to non-cancerous human cells, although are generally toxic to malignant cells. Still, on bismuth compounds as anticancer drugs, some studies have shown that bismuth drugs minimize the nephrotoxicity induced by cisplatin without affecting its antitumor effects [119,120]. Moreover, it was suggested that bismuth reduces cisplatin-induced nephrotoxicity by enhancing glutathione conjugation and vesicular transport [121]. Some studies suggest that combining bismuth and cisplatin may be a promising strategy for reducing cisplatin toxicity, but further research is necessary.

Several research groups worldwide have dedicated efforts to preparing bismuth compounds with potential anticancer properties. The studies show that low active or inactive free ligands generally enhance their anticancer activity upon bismuth coordination. The ligands can usually act as metal carriers increasing the solubility and bioavailability of Bi^3+^ [10], the agent responsible for the biological action. In addition, the outcomes have shown that the characteristics of the ligand might influence the anticancer activity of the resulting bismuth complexes, which are partly dependent on the type, the lipophilicity, and the electronic effects of the ligands. It is observed that bismuth complexes containing chelating ligands generally have superior cytotoxic activity against cancer cells compared to compounds bearing monodentate ligands. Thus, the additional thermodynamic stability in the physiological medium provided by the chelating nature of the ligands has a crucial role in the release rate of Bi ions and, consequently, in the biological action [52]. Still, regarding the role of the ligands in the anticancer activity of bismuth complexes, the presence of lipophilic groups on the molecular structure of the ligand may increase the uptake of target bismuth compounds, thereby enhancing the antiproliferative activity [122], as observed for lipophilic chelating thiol compounds [123,124,125,126]. In addition, the electronic effect and the dipole moment of bismuth-based compounds profoundly influence biological activity: polar compounds generally have better activity against cancer cells than non-polar ones [123,127]. Interestingly, the anticancer potential of bismuth compounds extends beyond bismuth complexes; organobismuth compounds have also been reported for their significant anticancer activities [128,129,130,131,132].

Although the mechanisms of antitumor action of bismuth compounds still need to be fully elucidated, some research indicates that cytotoxic pathways could be related to inducing apoptosis in tumor cells. The apoptosis process could be associated with the enhanced generation of intracellular reactive oxygen species (ROS), which are pro-oxidants and are critical in regulating cell death [129,130,131,133,134]. In addition, the cytotoxicity of bismuth could also be related to its capacity to inhibit some enzymes, such as proteases. In this sense, some bismuth compounds activated caspases and cell death by apoptosis [123,128,135,136,137]. A third route to induce apoptosis by bismuth-based compounds is a mitochondrial perturbation, i.e., a reduction of mitochondrial membrane potential [133,134,136]. Furthermore, Bi^3+^ complexes with aromatic heterocyclic ligands may trigger apoptosis through the DNA fragmentation pathway [83,132]. However, it is essential to mention that apoptosis is not the only regulated type of cell death. In recent decades, different signaling pathways and molecular mechanisms other than apoptosis have been detected, providing potential targets and new approaches for cancer treatment [138]. In this context, Iuchi and collaborators have demonstrated that some heterocyclic organobismuth compounds promote non-apoptotic cell death in vitro in human cancer cells via lipid peroxidation [139].

Considering the above discussions, some bismuth complexes that presented relevant anticancer activity investigated in the last decade are discussed below, and their values of an inhibitory concentration of 50% (IC_50_) on cancer cell lines are summarized in Table 2. It is relevant to highlight that many of these complexes were also evaluated for their antibacterial potential, and, generally, the complexes presented dual action as antibacterial and anticancer agents [140,141,142]. These findings are fascinating because microbial infection is one of the most common complications during cancer treatment due to surgical complications, chemotherapy- and radiotherapy-related neutropenia, and immunosuppressive drugs during cancer treatment [143]. In this way, developing drugs with dual action is greatly desired due to the increasing number of cancer patients and resistant microbial infections. In this sense, it is interesting to note that several antimicrobial agents, such as anthracyclines, mitomycin, and bleomycin, have been used to treat various cancers [144].

#### 3.4.1. Bismuth(III) Complexes Bearing S-Donor Ligands

Thiol chelation generally improves the solubility and lipophilicity of bismuth-based compounds [122], thus resulting in the potentiation of the antitumoral activity, as observed for antimicrobial activity. The greater thermodynamic stability of Bi–S compounds compared to Bi–O compounds and the relative Bi–S low lability give thiobismuth compounds broad biological applicability [20,32]. Since thiobismuth compounds can promptly exchange with free thiols due to a kinetically labile Bi–S bond, Bi^3+^ has high mobility inside cells, making it more suitable for reaching the desired biological targets.

Indeed, several Bi^3+^ complexes with bidentate *S*,*S*′-chelating ligands have shown relevant antitumor activity (Figure 8). For instance, Bi^3+^ dithiocarbamate complexes have demonstrated low toxicity to normal cell lines and remarkable cytotoxicity in various human carcinoma cancer cell lines [123,137,152]. In this context, Ishak and coworkers demonstrated that the bismuth diethyldithiocarbamate compound (**30**) is highly cytotoxic to hepatocellular carcinoma (HepG2), promotes apoptosis, blocks the cell cycle, and inhibits cell invasion [137]. Complementing these findings, recently, Chan et al. also explored the remarked antiproliferative activity of **30** on MCF-7 breast cancer cells, and further in vitro studies indicated that **30** induced apoptotic cell death through the intrinsic pathway and significantly inhibited MCF-7 cell invasion, which closely associated with angiogenesis and cancer metastasis [123]. Tamilvanan et al. reported three Bi^3+^ dithiocarbamate complexes containing furfuryl group (**31**–**33**) that displayed high antiproliferative activity against human cervix carcinoma KB (HeLa derivative) cell lines [124]. In other studies, Ozturk et al. reported mono- and dinuclear Bi^3+^ complexes with dithiocarbamate ligands (**34** and **35**) that exhibit expressive antiproliferative activity against MCF-7 and human cervix adenocarcinoma (HeLa) cell lines [125]. Also, the same research group prepared five Bi^3+^ halide dithiocarbamate complexes (**36**–**40**) that exhibit expressive activity on MCF-7 and HeLa cell lines with similar or even better activity than tamoxifen (an antiestrogen drug) [126]. Additionally, the most active complexes (**39** and **40**) could block the estrogen receptors on MCR-7 cells [126]. López-Cardoso and collaborators prepared four Bi^3+^ complexes of pyrrolidine-dithiocarbamate that presented antiproliferative effects against MCF-7 cells but were not significantly toxic to MDA-MB-231 breast cancer cell lines and showed some toxicity to healthy cells (MCF-10 A) [153].

In contrast, some Bi^3+^ compounds bearing S-monodentate ligands seem to generally show low cytotoxic values against cancer cells compared to compounds containing chelating ligands. For example, Ozturk et al. reported recently several Bi^3+^ coordinated in an S-monodentate fashion to thiosemicarbazone and other ligands, finding that most of them give no significative cytotoxicity against MCF-7 cells, IC_50_ (48 h) > 30 µM, and some of them presented moderate activity, IC_50_ (48 h) = 5.4–8.0 µM, against this cancer cell line [154,155,156,157]. Very recently, this research group reported five Bi^3+^ halide complexes with thiosemicarbazone bearing an acetylthiophene moiety that presented moderate cytotoxicity on HeLa cells (IC_50_ 4.5–24.3 µM) [158]. They also reported several Bi^3+^ complexes coordinated in an S-monodentate mode to thioamide ligands that showed low to moderate activity against human adenocarcinoma cells, HeLa (cervix) and MCF-7 (breast), with most IC_50_ (48 h) values superior to 30 µM [159,160]. Another Bi^3+^ compound bearing 6-mercaptopurine, an S-monodentate ligand, presents moderate cytotoxic activity on A549 and H460 human lung cancer cells with IC_50_ (24 h) ranging from 7 to 11 µM [161]. These outcomes indicated that chelating ligands are crucial for preparing Bi^3+^ complexes with high antitumoral activity.

#### 3.4.2. Bismuth(III) Complexes Bearing N,S-Donor Ligands

Several thiosemicarbazone Bi^3+^ complexes (Figure 9) were reported for their anticancer potential. For instance, two Bi^3+^ complexes containing the pentadentate *S*,*N*,*N*′,*N*′,*S*-chelating thiosemicarbazone derivative ligands (**41** and **42**) were reported [145,146]. Complex **41** exhibited cytotoxic activity against K562 leukemia cells and inhibited H22 xenograft tumor growth in tumor-bearing mice [146]. In the same direction, complex **42** greatly suppressed colony formation and migration and significantly induced the apoptosis of human lung cancer cells A549 and H460 but was not toxic to the non-cancerous human lung fibroblast (HLF) cell line [145]. Moreover, in vivo, **42** inhibited A549 xenograft tumor growth on tumor-bearing mice and did not indicate harmful effects on mouse weight and liver [145]. The same research group reported another Bi^3+^ thiosemicarbazone derivative complex in which the ligand binds to the metallic center in a bidentate *N*,*S*-chelating mode (**43**). The complex presented moderate cytotoxic activity on A549 and cytotoxic activity on H460 lung cancer cells that was higher than cisplatin. It also induced cell apoptosis and was not toxic to non-cancerous cells (HFL) [147].

Several Bi^3+^ complexes containing tridentate *N*,*N*′,*S*-chelating thiosemicarbazone derivative ligands with anticancer potential were reported. For example, two Bi^3+^ thiosemicarbazone complexes prepared by Bisceglie et al. (**44** and **45**) were evaluated against A549 human lung carcinoma cells and normal HuDe human epithelial tissue {Formatting Citation}. Both complexes were nontoxic to normal cells; **44** was highly selective for cancer cells and approximately 13-fold more active than cisplatin against A549 cancer cells, while **45** was almost inactive in this cell line [62]. A series of five Bi^3+^ thiosemicarbazone complexes reported by Lessa and coworkers presented moderate activity against glioblastoma multiforme SF-295 and colon adenocarcinoma HCT-116 human tumor cell lines [162]. Li et al. prepared four Bi^3+^ thiosemicarbazone complexes (**46**–**49**) that presented significative in vitro antiproliferative activity on HepG2 cells and promoted apoptosis in HepG2 cells associated with an increase in intracellular ROS production and a reduction of mitochondrial membrane potential (**47** and **48**) and caspase-3 activation (**47** and **49**) [133,135,148]. The four complexes have similar structures, but the difference in cytotoxic activities and apoptosis mechanisms indicate the relevance of the substituent groups’ ligands in the anticancer activity of the resulting complexes. Interestingly, two other Bi^3+^ complexes, reported by Li and collaborators, with similar structures to the above-cited complexes, present distinct antitumoral potential. Whereas the complex containing N(4)-pyridylthiosemicarbazone (**50**) presented high cytotoxic activity on K562 cells, IC_50_ (24 h) = 1.6 µM [149], the complex containing N(4)-phenylthiosemicarbazone presented just moderate cytotoxic activity under the same conditions (IC_50_ = 46.2 µM) [163].

#### 3.4.3. Bismuth(III) Complexes Bearing N,O-Donor Ligands

Although bismuth prefers S-donor ligands, it could form relatively stable complexes with N,O-donor chelating ligands, and some of them (Figure 10) presented relevant anticancer activity. For instance, Pereira-Maia et al. recently reported two new Bi^3+^ complexes with asymmetric *N*,*N*′,*O*-chelating ligands (**51** and **52**), highly cytotoxic against a chronic myelogenous leukemia cell line (K562), able to induce apoptosis [140]. In another interesting work, Beraldo and coworkers prepared five Bi^3+^ complexes containing *N*,*N*′,*O*-chelating hydrazones derivatives, which are stable in an aqueous solution. The complexes, **53**–**56**, were active on HL-60, Jurkat, and THP-1 leukemia, and MCF-7 and HCT-116 solid tumor cells [141]. Interestingly, these complexes were revealed to be as or more active than the reference drug chloramphenicol in several bacteria lines [141]. The same research group reported three Bi^3+^ complexes, **3**–**6**, Figure 4, bearing a planar macrocyclic ligand with expressive cytotoxic activity against several cancer cell lines. The mechanism of action of **3**–**6** seems to be related to the apoptotic pathway due to the DNA fragmentation induced by the complexes [83].

Considering the structural aspects in this case, the metal center, some dinuclear Bi^3+^ complexes with potential anticancer properties deserve attention. For example, Li et al. recently prepared seven dinuclear Bi^3+^ complexes bearing *N*,*O*,*N*-chelating Schiff-base ligands (**57**–**62**) that revealed significative cytotoxic activity on human gastric cancer SNU-16 cells [142,150].

#### 3.4.4. Organobismuth(III) Compounds

Iuchi and coworkers reported three heterocyclic organobismuth(III) compounds, **63–65**, Figure 11, with potent antiproliferative activity in vitro in various human cancer cell lines [136,164]. Compounds **63** and **64** exert potent antiproliferative activities in vitro in human cancer cell lines; **63** stimulated cell death in a ROS-dependent manner, inducing both apoptotic and nonapoptotic cell death [164], while **64** induced lipid peroxidation with the concomitant induction of caspase-independent cell death [139]. The eight-membered ring heterocyclic organobismuth(III) compounds (**63** and **64**) display higher antitumor activity than the six-membered ones (**65**) [122]. In this context, Liu and collaborators also reported eight-membered heterocyclic hypervalent organobismuth(III) compounds with moderate to high anticancer activity (**66–71**). The most active compound (**67**) has selective cytotoxicity against cancer cells, and the toxicological activity is related to the induction of apoptosis by caspase-3 activation [128]. Interestingly, **67** has the nitro substituted in the organogroup, showing the relevance of the ligand’s electronic effect in the metal-based compound’s cytotoxicity [165].

Some organobismuth(V) also presented significant anticancer potential; Figure 12 shows some examples. Onishi and collaborators tested the antitumor activity of tri(*p*-tolyl)bismuthane and a complex in which one *p*-tolyl ligand was replaced by phenyldiazenyl-pyrrolidine (**72**) [129]. While tri(*p*-tolyl)bismuthane exerted only a weak antiproliferative effect up to 10 μM, **72** exhibited high antiproliferative activity on several human cancer cell lines, indicating that the polar bismuth compound was better than the non-polar derivatives for biological applications [127]. In addition, treating human leukemia NB4 cells with **72** induced apoptosis, a loss of mitochondrial membrane potential, and the generation of ROS [129]. Interestingly, the same chemical structure of **72** with antimony in place of bismuth did not show any cytotoxic activity, confirming the antiproliferative effect of bismuth [129].

Demicheli et al. prepared two organobismuth(V) dicarboxylate complexes (**73** and **74**) and tested their antiproliferative activity on murine metastatic melanoma (B16F10) and chronic myelogenous leukemia (K562) cell lines as well as in healthy non-cancerous cell lines. Complex **73** was non-toxic and showed cytotoxic activity on K562 cells superior to cisplatin. Moreover, evaluating the pro-apoptotic activity of **73** in B16F10 cells indicates that cell cycle arrest and cell apoptosis contribute to drug cytotoxicity [130]. Another organobismuth(V) carboxylate complex (**75**) demonstrated significant antiproliferation activity on BT474 breast cancer cells, inducing an increased level of intracellular ROS, which leads to BT474 cell death through the apoptotic pathway [131]. Cui et al. prepared three organobismuth(V) carboxylate complexes, **76–78,** that present significative cytotoxic activity on MDA-MB-231 breast cancer cells and bound to CT-DNA via an intercalative mode [132].

### 3.5. Antiviral Activity

The emergence and reemergence of infectious diseases, especially those of viral origin, have concerned the scientific and medical communities in the last few years and caused global economic welfare challenges. For instance, the recent pandemic caused by coronavirus SARS-CoV-2, which has resulted in several million deaths worldwide, exposed research fragility for developing new antiviral agents. In addition, many neglected viral diseases, such as the Dengue, Zika, and Chikungunya viruses, require the development of effective antivirals. In this scenario, research for bismuth-based antiviral drugs can be a way out in this context of the critical demand for effective drugs to prevent and treat infectious diseases. In this sense, some works have demonstrated the potential of bismuth compounds for treating viral infections, such as rotavirus and HIV [166,167,168]. Moreover, two previous studies conducted by Sun and coworkers have demonstrated that several bismuth compounds could potentially inhibit the helicase enzymatic activities of coronavirus [169,170]. More specifically, it was shown that two of them were highly effective against SARS-CoV, inhibiting both coronavirus’s ATPase and helicase enzymes [170]. However, despite the promising results, only 14 years later, when COVID-19 became a global public health emergency, was interest in the findings of Sun and collaborators resumed.

Thus, in the last few years, the COVID-19 pandemic stimulated the development of new vaccines and drugs, and a few reports confirmed the potential anti-SARS-CoV-2 activity of Bi^3+^ compounds and provided valuable insight into the anti-SARS-CoV-2 mechanisms of bismuth drugs. The main molecular target of the investigations was the zinc-dependent SARS-CoV-2 non-structural protein 13 (nsp13). This protein, which pertains to the helicase superfamily 1B, in conjunction with the complex nsp7/nsp8/nsp12, plays a pivotal role in viral replication and the infection mechanism [171].

Recently, Sun and collaborators demonstrated that ranitidine bismuth citrate (RBC), a commercially available drug for treating duodenal ulcers, effectively prevents both the in vivo and in vitro replication of SARS-CoV-2 [17]. Furthermore, the affinity of bismuth to infected cells was high, with a selectivity index of 975. The authors also evidenced in in vitro studies that the inhibition of SARS-CoV-2 helicase by bismuth-based compounds can be attributed to the irreversible displacement of zinc(II) ions in SARS-CoV-2 nsp13 by Bi^3+^ ions [17]. The protein domains of SARS-CoV-2 nsp13 have a cysteine-rich region [172], which is an attractive molecular target for Bi^3+^ ions since they are known to have a high affinity for thiolate sulfur by forming Bi–S bonds [173]. Sun’s research group also showed that administering bismuth drugs combined with N-acetyl cysteine (NAC) constitutes a broad-spectrum anti-coronavirus cocktail therapy [167]. NAC further stabilized bismuth in stomach-like conditions, owing to the formation of a stable bismuth thiolate complex [Bi(NAC)_3_], enhancing bismuth drug uptake in tissues. Subsequently, Bi^3+^ ions suppressed virus replication by inhibiting multiple essential viral enzymes [174].

In 2020, Zhou and coworkers extended the comprehension of the biochemical characteristics of two replicative enzymatic activities associated with SARS-CoV-2 nsp13, nucleoside triphosphate hydrolase (NTPase) and RNA helix unwinding, and demonstrated that bismuth compounds inhibit their activities in a dose-dependent manner [18]. In 2021, Tao and collaborators, in turn, showed that an essential SARS-CoV-2 protease, 3CLpro (3-chymotrypsin-like protease), is one of the targets for colloidal bismuth subcitrate (CBS), a commercially available drug. Furthermore, they demonstrated that Bi^3+^ first bound to the metal site at the C-terminal domain of 3CLpro and caused dimeric 3CLpro to dissociate into monomers. This alteration resulted in protease dysfunction, suppressing SARS-CoV-2 viral replication in the mammalian cell [16].

More recently, Marrone et al., employing a multilevel computational approach, provide further insight into the antiviral mechanism of bismuth drugs. They demonstrated that the alterations in nsp13 resulting from the process of Zn^2+^ substitution with Bi^3+^ generated significative non-local structural impacts [175]. This theoretical study indicated that the Zn^2+^ to Bi^3+^ exchange leads to substantial changes in the geometrical and ionization properties of nsp13, which decrease the interaction between nsp13 and nsp12, thus impacting the process of viral replication and the infection mechanism of SARS-CoV-2 [175].

Although metal ions can bind on single and often functionally crucial protein residues, the antiviral activities of metal-based compounds have rarely been explored [176,177]. The efficacy of metallo-antiviral agents is expected to be less liable to new viral variants, thus resulting in an enormous potential for treating viral infections. One attractive strategy employed in some of these recent reports was investigating the possibility of repurposing the old metallodrugs, RBC and CBS (Table 1), as therapies for treating COVID-19. However, it is noteworthy that the required concentrations of drugs for suppressed SARS-CoV-2 viral replication in vivo and cellular-based experiments are much higher than that in a proteolytic assay, probably due to the multiple-targeting property of bismuth drugs [16] and their low bioavailability. This highlights the need for designing more specific bismuth-based compounds for antiviral activity.

## 4. Strategies to Enhance the Bioavailability of Bismuth beyond the Complex Formation

As bismuth-based compounds easily suffer hydrolysis in physiological conditions, in addition to research for obtaining new bismuth complexes that are more bioavailable, approaches for the targeting and controlled delivery of bismuth-based compounds could be promising strategies for enhancing their bioavailability and, consequently, their efficacy.

### 4.1. Incorporation of Bismuth in Biopolymeric Systems

Polysaccharides are biopolymers that attract interest as promising drug delivery carriers due to their unique properties, such as biocompatibility, non-toxicity, biodegradability, and high availability [178]. The anion polysaccharides could act as suitable ligands for Bi^3+^, and the complex formed could reduce bismuth intake and enhance Bi efficacy. In this context, Zhu et al. prepared Bi^3+^–*Hericium erinaceus* polysaccharide complexes that significantly inhibited *H. pylori* (MIC = 20 μg mL^−1^), even with a lower content of Bi^3+^ than the standard drug CBS [179]. Silva-Caldeira and coworkers also developed a biopolymeric system incorporating Bi^3+^ ions and the antibiotic furazolidone on an alginate–carboxymethylcellulose blend, obtaining a system able to release these therapeutic agents in a concomitant and controlled way that could be adequate for *H. pylori* treatment [180]. Jiang et al., on the other hand, prepared pectin–bismuth complexes and investigated their structural changes in an acidic environment. The authors observed that Bi^3+^ gradually dissociated from carboxyl groups in the acidic environment, which is a beneficial property for treating *H. pylori* infections [181]. Andrews et al. proceeded with incorporating Bi compounds into microfibrillated cellulose producing composites that display significative antibacterial activity at low complex loadings, with the advantage of controlling the leaching of the Bi complexes from cellulose composites through structural changes in Bi compounds [57,58].

Aiming to obtain a biocompatible antimicrobial hydrogel for active wound dressing applications, Maliha et al. recently prepared a nanocellulose hydrogel incorporated with a phenyl bis-phosphinato Bi^3+^ complex. The prepared hydrogel showed bactericidal activity against *Acinetobacter baumannii* and *Pseudomonas aeruginosa* and a bacteriostatic effect against antibiotic-resistant bacteria MRSA and VRE, while having no toxic impact on mammalian fibroblast cells, thus revealing that bismuth-based compounds could act as a safe antimicrobial additive for preparing materials for biomedical applications [182].

Administering bismuth from polysaccharides is a good strategy for improving Bi bioavailability and diminishing the dose needed to reach the desired therapeutic outcome. In addition, the outcomes evidence the great potential of polysaccharides incorporated with Bi compounds for many applications, including antibacterial surfaces and materials. Even though there are few reports in the literature, the potentialities of biopolymers containing Bi^3+^ or Bi-based compounds for treating bacterial infections could open a new research trend. In addition, in recent years, nanotechnology-based approaches for targeting and controlling delivery, enhancing bioavailability, and minimizing systemic toxicities of metal-based therapeutic agents have revealed significant potential [183]. In this context, polymer-based nanosized systems designed to load physically or covalently conjugate Bi compounds could also be an interesting strategy for improving their bioavailability and, consequently, their absorption rate. However, it should be noted that, to the best of our knowledge, no research group has yet employed polymeric nanocarriers for bismuth-based agents [183]. In contrast, research on medical applications of metallic bismuth nanoparticles has grown tremendously in recent years [184].

### 4.2. Bismuth-Based MOFs and Coordination Polymers

Concerning the high porosity and surface area, as well as the infinite building blocks that can be combined to form a variety of materials, metal–organic frameworks (MOFs) are relevant in several fields, including pharmacological purposes [185]. These promising multifunctional platforms are suitable as drug delivery systems because they can accommodate small molecules, act as drug storage, and have low cytotoxicity, high biodegradability, and high biocompatibility [186]. Indeed, some bismuth-based MOFs were recently prepared as drug carriers since bismuth and its compounds are commonly considered biologically safe and nontoxic [187,188]. Interestingly, Huang and coworkers recently prepared two bismuth-based MOFs to act as a reservoir and controlled the delivery of Bi^3+^ ions to extend the antimicrobial duration, shedding light on a new administration modality of bismuth. In addition, the Bi-MOFs presented low cytotoxicity in human gingival fibroblasts, as expected, and exhibited selective antimicrobial activities against Gram-negative oral pathogens, *A. actinomycetemcomitans* (MIC = 7.8–31.3 µg mL^−1^) and *P. gingivalis* (MIC = 7.8–62.5 µg mL^−1^), in planktonic and biofilm forms [189].

Burrows et al. prepared some bismuth coordination networks containing deferiprone, an iron chelator drug, aiming to obtain a compound with a potential dual mode of action: simultaneously a combination of a system for deferiprone controlled release and a system with antibacterial activity. The compounds exhibit appreciable *H. pylori* inhibitory action, comparable to that of BSS, and could release the molecule of interest [190].

Bismuth has immense importance in material chemistry due to its unique properties. Indeed, numerous bismuth-based materials with varied biomedical applications, such as treating cancer and microbial infections, diagnostics imaging, theranostics, and biosensing, have recently been developed [191]. Some of these nanoparticle materials were designed to be administered directly into the bloodstream; therefore, the fates and toxicity of the Bi liberated from the nanoparticles need further research and the additional development of systems to control the release of bismuth.

## 5. Conclusions and Outlook

The medicinal chemistry of bismuth presents numerous opportunities to address significant health concerns. One of the most pressing global health challenges is antimicrobial resistance, exacerbated by the lack of new and effective antibiotics. Bismuth’s low toxicity in human cells and its ability to combat several microorganisms, including resistant bacteria strains, make it a promising candidate for further exploration. To develop more effective bismuth drugs, it is crucial to find ways to increase the bioavailability of this metal ion in the body’s physiological environment. Preparing and co-administering bioavailable bismuth agents with antibiotics and developing devices to simultaneously release both bismuth and antibiotics in a controlled manner are methods that can repurpose existing antibiotics for clinical use and overcome antimicrobial resistance.

It is also worth noting that various bismuth complexes have demonstrated notable anticancer properties. Interestingly, many of these compounds also possess antimicrobial effects. Developing a drug that can combat both cancer and bacterial infections simultaneously is a forward-thinking step in cancer treatment, as hard-to-treat microbial infections are one of the most common hurdles during treatment.

Another two points to highlight about the medicinal chemistry of bismuth are its antiviral and anti-leishmanial potential. The significant antiviral activities demonstrated by a few bismuth compounds suggest that a large research field can still be explored, which could help develop effective antivirals, especially for neglected diseases caused by viruses. In addition, bismuth’s anti-leishmanial potential makes it a safer alternative to the antimony-based compounds currently used to treat the leishmaniasis, an NTD, indicating reasonable scope for further research on developing bismuth-based anti-leishmanial agents.

Finally, continuous research in the field is necessary because bismuth formulations currently used as pharmaceuticals face insolubility, low absorption, and difficulties with synthetic reproducibility. Developing new bismuth-based agents requires a delicate balance of properties for stability, solubility, targeting, and subsequent activity. In this context, as highlighted in this review, controlled delivery systems for incorporating bismuth agents could be an interesting strategy for enhancing the bioavailability of active bismuth-based compounds.

## Figures and Tables

**Figure 1 molecules-28-05921-f001:**
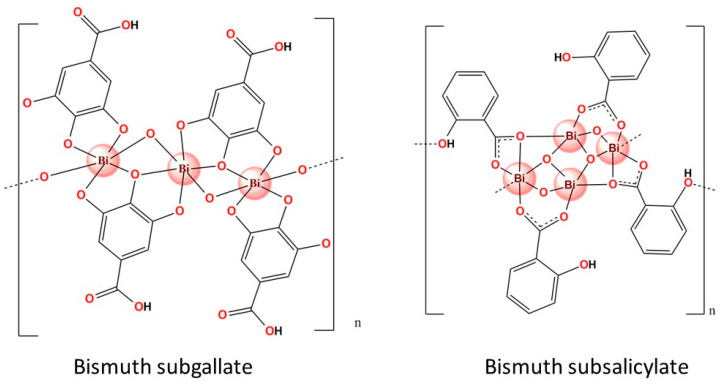
Determined structures of two bismuth-containing pharmaceuticals [4,5].

**Figure 2 molecules-28-05921-f002:**
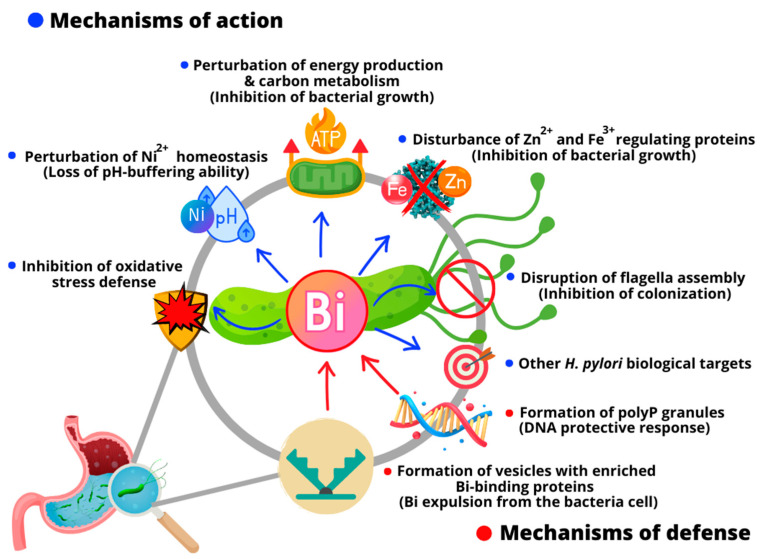
Schematic representation of the main action mechanisms of bismuth against *H. pylori* and probable bacterial resistance mechanisms.

**Figure 3 molecules-28-05921-f003:**
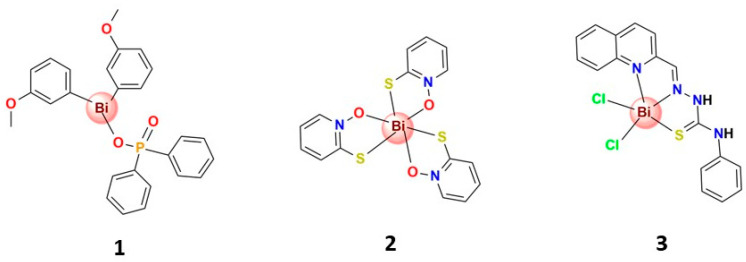
Structures of some bismuth(III) complexes that present significant antibacterial activity.

**Figure 4 molecules-28-05921-f004:**
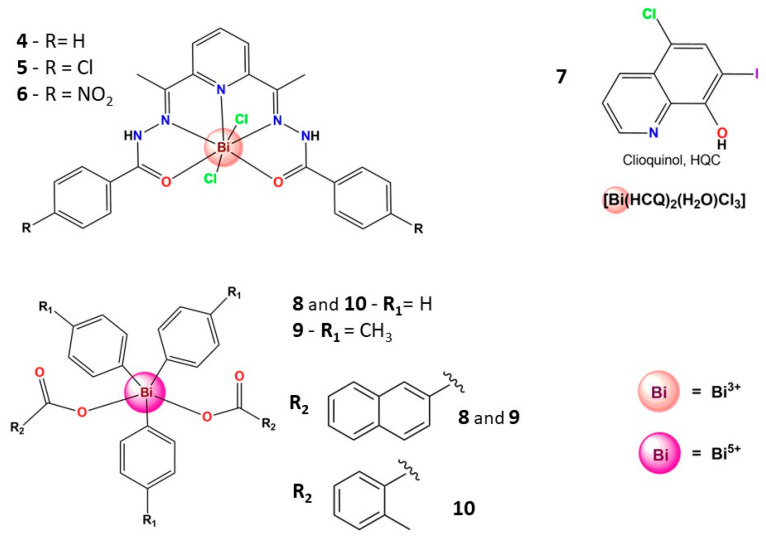
Structures of some bismuth(III/V) complexes that present antifungal activity.

**Figure 5 molecules-28-05921-f005:**
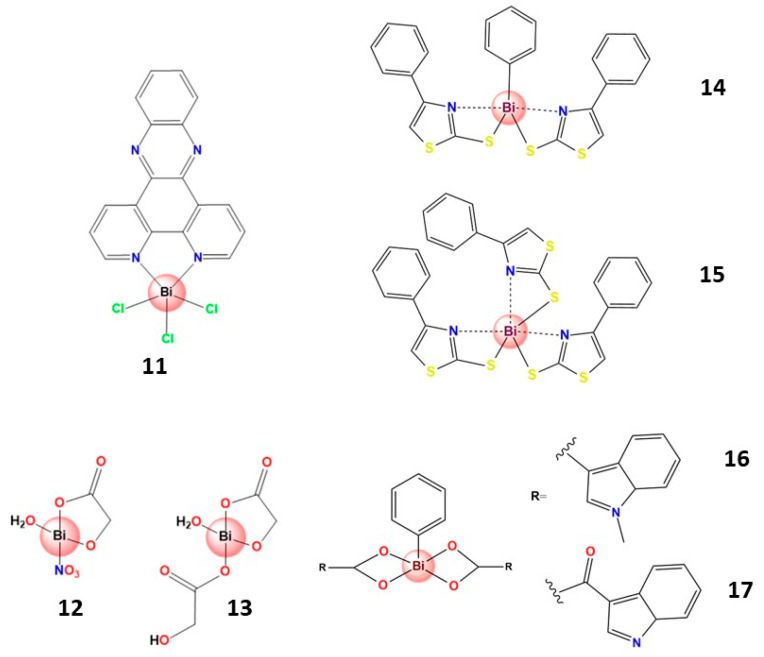
Structures of some bismuth(III) complexes that present significative anti-leishmaniasis activity.

**Figure 6 molecules-28-05921-f006:**
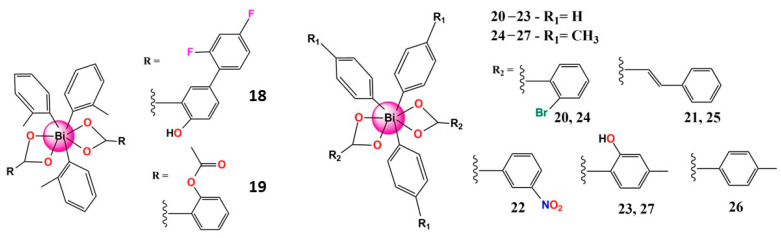
Structures of some bismuth(V) complexes that present significative anti-leishmaniasis activity.

**Figure 7 molecules-28-05921-f007:**
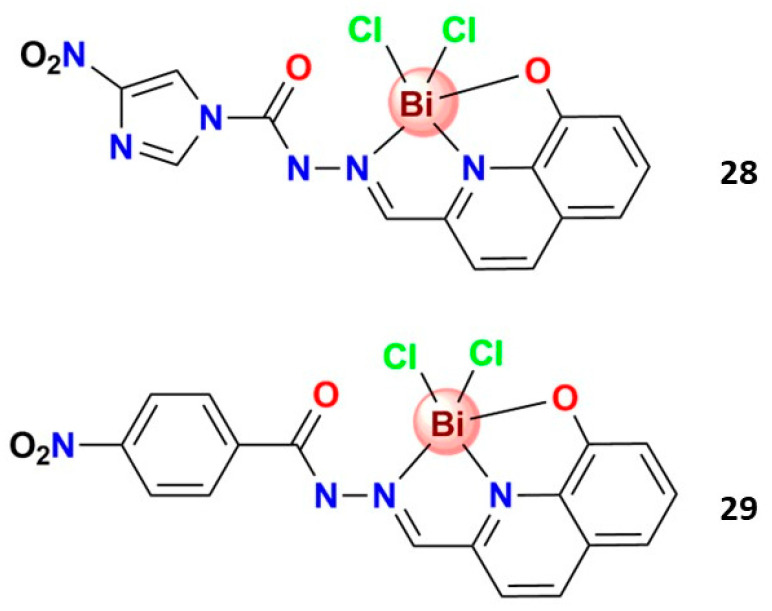
Structures of bismuth(III) complexes with anti-*Trypanosoma cruzi* activity.

**Figure 8 molecules-28-05921-f008:**
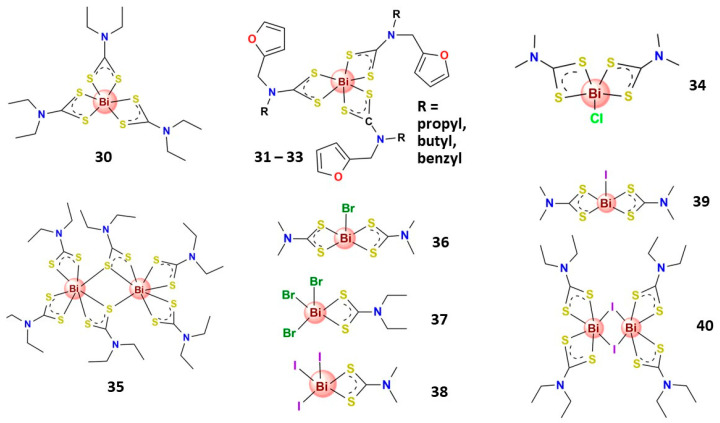
Structures of some bismuth(III) complexes bearing S-donor ligands that present significative anticancer activity.

**Figure 9 molecules-28-05921-f009:**
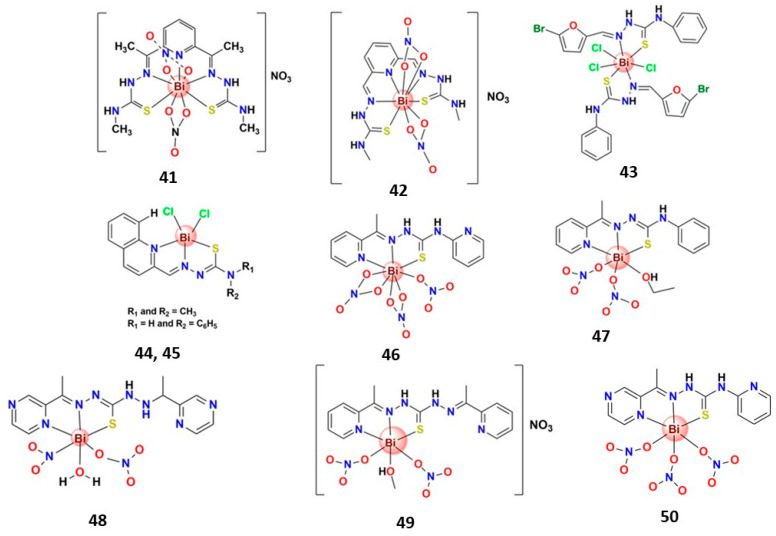
Structures of some bismuth(III) complexes bearing N,S-donor ligands that present significative anticancer activity.

**Figure 10 molecules-28-05921-f010:**
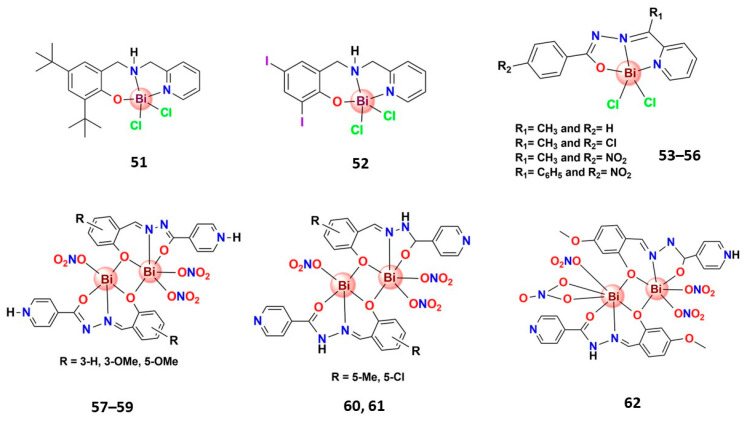
Structures of some bismuth(III) complexes bearing N,O-donor ligands that present significative anticancer activity.

**Figure 11 molecules-28-05921-f011:**
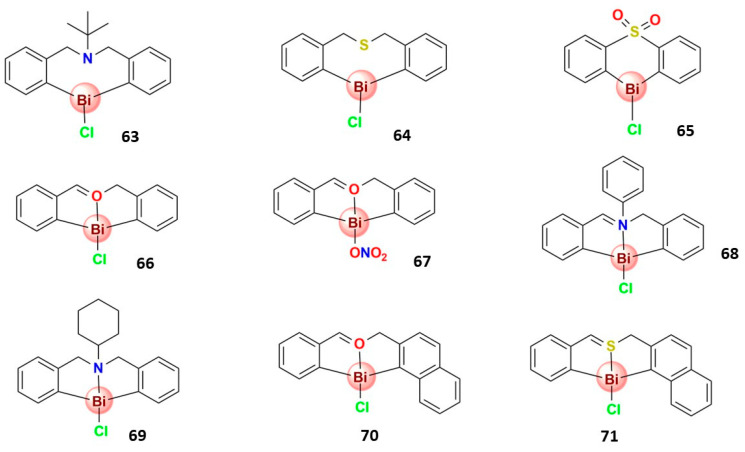
Structures of some heterocyclic organobismuth(III) compounds that present significative anticancer activity.

**Figure 12 molecules-28-05921-f012:**
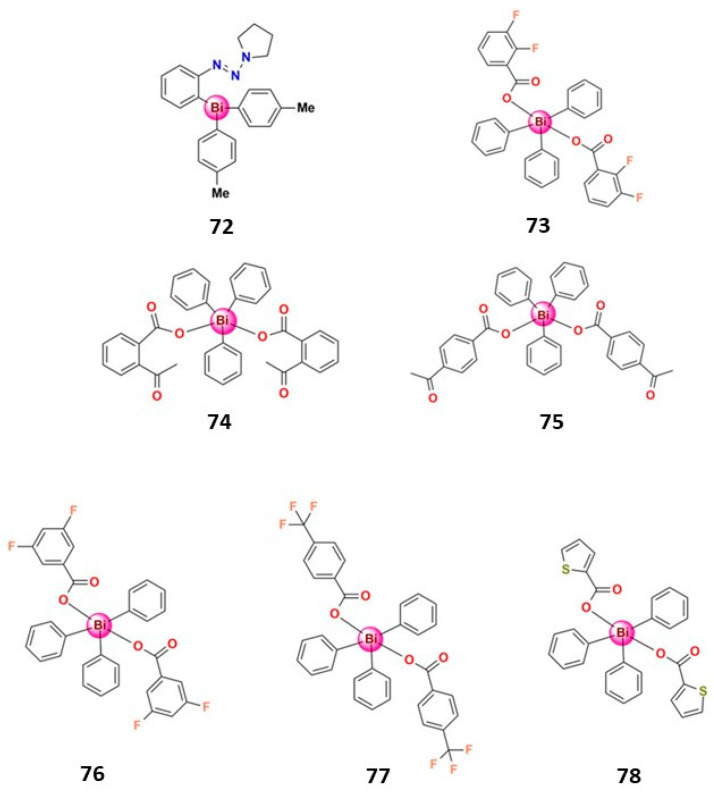
Structures of some organobismuth(V) compounds that present significative anticancer activity.

**Table 1 molecules-28-05921-t001:** Clinical uses of some bismuth compounds.

Bismuth Compound	Brand Name	Clinical Use
Bismuth subgallate	–	Improving stool consistency and odor in colostomy and ileostomy patients
Bismuth oxide	–	Wound infection
Bismuth subnitrate	–	Irritable colon, gastric disorders, constipation
Bismuth phosphate, aluminate, and subcarbonate	–	Several gastrointestinal disorders
Sodium bismuth tartrate	–	Syphilis and canker sores
Colloidal bismuth subcitrate (CBS)	De-Nol^®^	Gastric and duodenal ulcers, non-ulcer dyspepsia, *H. pylori*
Bismuth subsalicylate (BSS)	Pepto-Bismol^®^	Dyspepsia, diarrhea, *H. pylori*
Ranitidine bismuth citrate (RBC)	Tritec^®^, Pylorid^®^	Gastric and duodenal ulcers, *H. pylori*
Tribromophenatobismuth(III)	Xeroform^®^	Antibiotic in wound dressings

Data from references [3,6,7,8].

**Table 2 molecules-28-05921-t002:** IC_50_ values (µM) from several bismuth compounds on cancer cell lines for incubation times of 12 h (^ᴥ^), 24 h (*), 48 h (^§^), or 72 h (^†^).

Compounds	HepG2	MCF-7	HeLa	K562	A549	H460	B16F10	BT474	MDA-MB-231	SNU-16	HCT116	HL60	THP-1	NB4	Refs.
**30**	0.5 *	1.26 *	–	–	–	–	–	–	–	–	–	–	–	–	[137]
**31–33**	–	–	0.40 *–0.95 *	–	–	–	–	–	–	–	–	–	–	–	[124]
**34, 35**	–	0.023 ^§^; 0.043 ^§^	0.33 ^§^; 0.19 ^§^	–	–	–	–	–	–	–	–	–	–	–	[125]
**36–40**	–	0.070 ^§^–0.100 ^§^	0.05 ^§^–0.30 ^§^	–	–	–	–	–	–	–	–	–	–	–	[126]
**41**	–	–	–	–	3.2 *	3.6 *	–	–	–	–	–	–	–	–	[145]
**42**	–	–	–	26.8 *	–	–	–	–	–	–	–	–	–	–	[146]
**43**	–	–	–	–	16.4 *	20.0 *	–	–	–	–	–	–	–	–	[147]
**44, 45**	–	–	–	–	5.0 *	–	–	–	–	–	–	–	–	–	[62]
**46, 47**	1.6 *, 3.4 *	–	2.7 *, 9.0 *	1.8 *, 5.2 *	–	–	–	–	–	–	1.6 *, 5.57 *	–	–	–	[134,148]
**48, 49**	2.96 *, 3.42 *	–	–	–	–	–	–	–	–	–	–	–	–	–	[133,135]
**50**	–	–	–	1.6 *	–	–	–	–	–	–	–	–	–	–	[149]
**51, 52**	–	–	–	0.30 ^†^, 0.38 ^†^	–	–	–	–	–	–	–	–	–	–	[140]
**53–56**	–	0.23 ^§^–10 ^§^	–	–	–	–	–	–	–	–	1.42 ^§^–4.47 ^§^	0.20 ^§^–1.18 ^§^	0.43 ^§^–2.78 ^§^	–	[141]
**3–6**	–	0.27 ^§^–1.07 ^§^	–	–	–	–	–	–	–	–	2.83 ^§^–10.22 ^§^	0.09 ^§^–0.23 ^§^	–	–	[83]
**57–62**	–	–	–	–	–	–	–	–	–	0.3–1.6 ^†^	–	–	–	–	[142,150]
**63–65**	–	–	4.27–4.85 ^ᴥ^	0.69–0.95 ^ᴥ^	1.15–4.29 ^ᴥ^	–	–	–	–	–	–	0.15–1.13 ^ᴥ^	–	0.05–0.25 ^ᴥ^	[136]
**66–71**	–	–	–	–	2.2 *–26.7 *; 1.8 ^§^–18.6 ^§^; 0.8 ^†^–12.6 ^†^	–	–	–	–	–	–	–	–	–	[128]
**72**	–	–	5.36 ^ᴥ^	3.33 ^ᴥ^	3.34 ^ᴥ^	–	–	–	–	–	–	1.44 ^ᴥ^	–	0.88 ^ᴥ^	[129]
**73, 74**	–	–	–	3.0 ^†^; 19.6 ^†^	–	–	11.9 ^†^; 11.4 ^†^	–	–	–	–	–	–	–	[130]
**75**	–	–	–	–	–	–	–	3.9 *	–	–	–	–	–	–	[131]
**76–78**	–	–	–	–	–	–	–	–	<20 *	–	–	–	–	–	[132]
CDDP	152.0 *	25.9 *; 5.5 ^§^–6.8 ^§^	10.0 ^§^	1.10 ^†^	(23.8–39.8) *; 10.7 ^§^; 9.8 ^†^	46.2 *	–	–	–	–	>100 ^§^	0.05 ^§^	>100 ^§^	–	[125,128,137,141,151]
DOX	4.6 *	0.018 ^§^	1.12 ^§^	–	–	–	–	–	–	0.23 ^†^	–	–	–	–	[125,137,142]
TMX	–	0.046 ^§^	–	–	–	–	–	–	–	–	–	–	–	–	[126]
MIT	5.3 *		5.98 *	–	–	–	–	–	–	–	8.34 *	–	–	–	[148]

Cisplatin (CDDP), Doxorubicin (DOX), Tamoxifen (TMX), and Mitoxantrone (MIT).

## Data Availability

Not applicable.

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
