# Peer review of "Biological Activities of Bismuth Compounds: An Overview of the New Findings and the Old Challenges Not Yet Overcome"

_molecules, 2023, doi:10.3390/molecules28155921_

Round 1

Reviewer 1 Report

This review focuses on the most recent advancements in the field, providing valuable insights into the potential applications of bismuth complexes in medicinal chemistry. Notably, the procedures for synthesizing these bismuth(III) complexes from an organometallic perspective are currently lacking. Therefore, it would be beneficial to include some of these synthetic methods to enhance the comprehensiveness of the manuscript. Overall, I believe that with minor revisions, the manuscript fulfills all the requirements for publication in the journal 'Molecules.' Additionally, I have included additional suggestions and comments for further improvements:  

(1) See the abstract section, line 18. The authors discuss “on recent updates. However, it is very important to specify the range of years for the manuscripts included in this review.

(2) See lines 108 and 184. “H. pylori” should be written in italic.

(3) See 3.1.2. Antibacterial activity beyond H. pylori and reversion of bacterial drug resistance. I recommend including a Figure displaying the structures of some complexes with remarkable biological activity.

(4) See lines 369 to 378. The most significant IC50 values might be included.

(5) See lines 416 to 418. It would be advantageous to include the key values related to the biological activity and its standard drug.

(6) See Table 2. I recommend including the standard deviations corresponding to the reported IC50 values. The specific values should be assigned for each compound instead of using a range (i.e. 28-30). ¿Could you please provide the IC50 values for compound 28, 29, and 30?.

(7) See lines 498 to 520. Although the values of IC50 of compound are already reported in Table 2, I strongly suggest including these values in the discussion within this paragraph.

(8) See lines 525 to 527. The authors mention that “Another Bi3+ compound bearing 6-mercaptopurine, an S-monodentate ligand, presents moderate cytotoxic activity on human lung cancer cells”.  However, it would be beneficial to include the values for this activity. Otherwise, readers may need to search for them in the corresponding manuscript rather than finding them in this review.

(9) See lines 533 to 562. Despite the IC50 values of compound being reported in Table 2, I strongly suggest including these values in the discussion within these paragraphs.

(10) See lines 567 to 585. Despite the IC50 values of compound being reported in Table 2, I strongly suggest including these values in the discussion within these paragraph

(11) See lines 591-604 and 608-630. Despite the IC50 values of compound being reported in Table 2, I strongly suggest including these values in the discussion within these paragraphs. Also, information related with the mechanism of action reported in their manuscripts might be deepen in this review.

(12) See 4.1. Incorporation of bismuth in biopolymeric systems. It would be beneficial to include a figure illustrating the structures of these complexes. Additionally, specifying the values of their biological activity would provide further clarity.

(13) 4.2. Bismuth-based MOFs and coordination polymers. It would be beneficial to include a figure illustrating the structures of these complexes. Additionally, specifying the values of their biological activity would provide further clarity.

(14) See the reference section. The DOI of each artichle is required.

Reviewer 2 Report

This review paper on the Biological Activities of Bismuth Compounds, is a very interesting read and I personally will be using it a reference/guide in my research. The paper is well structured and the scope covers a number of important topics in this research area.

There are a few minor issues that should be addressed by the authors:

1. Figure 1 - structure of bismuth subgallate

2. Figure 2 - mechanism of defese defense

3. Table 2 - no entries for SF295 column

4. Figure 8 - structure of compounds 44 and 46

The language could be improved. There are a few language issues in the manuscript which could easily be resolved. Some (not all) will be listed below:

pg 2, line 46 - is a boiling field

pg 3, line 100 - Around the world

pg 7, line 694 - RBC e CDS

Reviewer 3 Report

This study is a review manuscript that includes the biological activities of bismuth compounds and that I believe can be a guide for those who will work in this field.

In the manuscript, the authors listed the bismuth-containing compounds according to their pharmacological properties.

I believe that the following additions by the authors to this review manuscript will make the article more comprehensive.

-While the authors mentioned Bismuth(III) complexes with S-donor ligands, they did not mention bismuth(III) thioamide complexes in this section. This part can add bismuth(III) thioamide complexes.

Inorganica Chimica Acta 497 (2019) 119094, doi.org/10.1016/j.ica.2019.119094

Main Group Met. chem. 2018; 41(5–6): 143–154, doi.org/10.1515/mgmc-2018-0035; et al.

I believe there are some missing references regarding bismuth(III) thiosemicarbazone complexes.

European Journal of Medicinal Chemistry 182 (2019) 111616, doi.org/10.1016/j.ejmech.2019.111616

New J. Chem., 2023, 47, 12779, DOI: 10.1039/d3nj01411h

Inorganic Chemistry Communications 20 (2012) 37–40, doi:10.1016/j.inoche.2012.02.009

Chemistry & Biodiversity – Vol. 9 (2012), 1955-1966, doi: 10.1002/cbdv.201100447

Dalton Trans., 2012, 41, 12882, DOI: 10.1039/c2dt31256e

Bioorganic & Medicinal Chemistry Letters 22 (2012) 2418–2423, doi:10.1016/j.bmcl.2012.02.024

The quality of the English language of the article is sufficient, it is useful to revise it for some minor mistakes.
